# Type 1 polyisoprenoid diphosphate phosphatase modulates geranylgeranyl-mediated control of HMG CoA reductase and UBIAD1

**Rania Elsabrouty, Youngah Jo, Seonghwan Hwang, Dong-Jae Jun, Russell A DeBose-Boyd***

Department of Molecular Genetics, University of Texas Southwestern Medical Center at Dallas, Dallas, United States

**Abstract** UbiA prenyltransferase domain-containing protein-1 (UBIAD1) utilizes geranylgeranyl pyrophosphate (GGpp) to synthesize the vitamin $K_2$ subtype menaquinone-4. The prenyltransferase has emerged as a key regulator of sterol-accelerated, endoplasmic reticulum (ER)-associated degradation (ERAD) of HMG CoA reductase, the rate-limiting enzyme in synthesis of cholesterol and nonsterol isoprenoids including GGpp. Sterols induce binding of UBIAD1 to reductase, inhibiting its ERAD. Geranylgeraniol (GGOH), the alcohol derivative of GGpp, disrupts this binding and thereby stimulates ERAD of reductase and translocation of UBIAD1 to Golgi. We now show that overexpression of Type 1 polyisoprenoid diphosphate phosphatase (PDP1), which dephosphorylates GGpp and other isoprenyl pyrophosphates to corresponding isoprenols, abolishes protein geranylgeranylation as well as GGOH-induced ERAD of reductase and Golgi transport of UBIAD1. Conversely, these reactions are enhanced in the absence of PDP1. Our findings indicate PDP1-mediated hydrolysis of GGpp significantly contributes to a feedback mechanism that maintains optimal intracellular levels of the nonsterol isoprenoid.

*For correspondence:
russell.debose-boyd@
utsouthwestern.edu

**Competing interest:** The authors declare that no competing interests exist.

## Editor's evaluation

This manuscript investigates the regulation of the mevalonate pathway by geranylgeraniol. In a series of elegant and convincing experiments, the authors show that modulating the expression levels of Polyisoprenoid Diphosphate Phosphatase (PDP1) alters endoplasmic reticulum-associated protein degradation of HMGCoA reductase though UBAID1. They also show that modulation of geranylgeraniol levels through exogenous addition or depletion of PDP1 alters stability and levels of important small GTPases, such as RhoA. The biology here is fascinating and very important, not only as it is pertinent to a fundamental mechanism of biological control but also as directly related to cholesterol-lowering statin therapy.

## Introduction

Mevalonate is a crucial intermediate in a branched pathway that produces cholesterol and nonsterol isoprenoids such as farnesyl pyrophosphate (Fpp), geranylgeranyl pyrophosphate (GGpp), ubiquinone-10, dolichol, heme, and the vitamin $K_2$ subtype menaquinone-4 (MK-4). These end products of mevalonate metabolism play essential roles in a variety of cellular processes ranging from the maintenance of membrane structure (cholesterol), protein prenylation (Fpp and GGpp) and asparagine-linked glycosylation (dolichol) to electron transport (ubiquinone-10 and heme) and γ-carboxylation of

glutamate residues in specific proteins (MK-4) (**Goldstein and Brown, 1990**; **Wang and Casey, 2016**). Dysregulation of mevalonate metabolism has been implicated in numerous diseases including certain types of cancers, Alzheimer's disease, osteoporosis, and non-alcoholic fatty liver disease (**Fusaro et al., 2017**; **Jeong et al., 2018**; **Mullen et al., 2016**; **Zhao et al., 2020**). Moreover, inhibition of mevalonate synthesis appears to elicit immunomodulatory and anti-inflammatory responses that influence infectious diseases caused by various pathogens (viruses, protozoa, fungi, and bacteria) (**Parihar et al., 2019**). Elucidation of mechanisms governing regulation of mevalonate metabolism is imperative for the understanding of disease pathology and development of attenuating therapies.

Synthesis of mevalonate is catalyzed by the enzyme 3-hydroxy-3-methylglutaryl coenzyme A (HMG CoA) reductase, which is anchored to the endoplasmic reticulum (ER) through an N-terminal membrane domain that precedes a cytosolic catalytic domain (**Liscum et al., 1985**; **Roitelman et al., 1992**). The reductase is targeted by a unique feedback regulatory system that operates through multiple mechanisms to ensure cells constantly produce nonsterol isoprenoids, but avoid overaccumulation of cholesterol and/or its sterol precursors (**Brown and Goldstein, 1980**). One of these mechanisms involves accelerated ER-associated degradation (ERAD) of reductase from membranes. Accumulation of certain sterols within membranes of the ER initiates ERAD of reductase by causing it to bind ER membrane proteins called Insig-1 and Insig-2 (**Sever et al., 2003a**; **Sever et al., 2003b**). Subsequent ubiquitination of reductase by Insig-associated ubiquitin ligases triggers its extraction across the ER membrane, followed by dislocation into the cytosol for degradation by 26 S proteasomes (**Elsabrouty et al., 2013**; **Jiang et al., 2018**; **Jo et al., 2011**; **Morris et al., 2014**; **Song et al., 2005**). The physiological significance of sterol-accelerated ERAD of reductase is highlighted by the finding that knock-in mice expressing endogenous reductase resistant to sterol-induced ubiquitination accumulate the protein in multiple tissues, despite reduced levels of its mRNA (**Hwang et al., 2016**).

Nonsterol, mevalonate-derived products augment the ERAD of reductase, but only in the presence of sterols (**Correll and Edwards, 1994**; **Nakanishi et al., 1988**; **Roitelman and Simoni, 1992**). The addition to cells of geranylgeraniol (GGOH), but not farnesol (FOH), reproduces the augmentation of sterol-accelerated reductase ERAD (**Sever et al., 2003a**). FOH and GGOH are alcohol derivatives of Fpp and GGpp, respectively. Subsequent studies revealed that GGOH synergizes with sterols to accelerate ERAD of reductase by enhancing extraction of the ubiquitinated enzyme across ER membranes (**Morris et al., 2014**). Important insight into GGOH-enhanced ERAD of reductase has been provided by the emergence of a new player in the pathway. This new player, UbiA prenyltransferase domain-containing protein-1 (UBIAD1), produces MK-4 by using GGpp to prenylate the naphthoquinone menadione (vitamin $K_3$) that is derived from dietary phylloquinone (vitamin $K_1$) (**Hirota et al., 2013**; **Nakagawa et al., 2010**). Current evidence indicates that sterols also stimulate binding of UBIAD1 to a subset of reductase molecules, inhibiting a post-ubiquitination step in their ERAD (**Schumacher et al., 2015**). Inhibition of reductase ERAD permits continuous synthesis of mevalonate for incorporation into GGpp and other nonsterol isoprenoids, even when sterols are abundant (**Schumacher et al., 2018**). The addition to cells of GGOH causes release of UBIAD1 from reductase; this release relieves inhibition of ERAD and triggers translocation of UBIAD1 from the ER to Golgi (**Schumacher et al., 2015**; **Schumacher et al., 2016**). Missense mutations in the *UBIAD1* gene cause Schnyder corneal dystrophy (SCD), an autosomal dominant eye disease in humans characterized by the corneal accumulation of cholesterol (**Orr et al., 2007**; **Weiss et al., 2007**). SCD-associated UBIAD1 resists GGOH-induced release from HMGCR and is sequestered in membranes of the ER (**Schumacher et al., 2015**; **Schumacher et al., 2016**). ERAD of reductase becomes inhibited by the ER sequestered, SCD-associated UBIAD1, which leads to enhanced synthesis and intracellular accumulation of cholesterol in both cultured cells and tissues of knock-in mice that express SCD-associated UBIAD1 (**Jo et al., 2019**; **Schumacher et al., 2018**).

Metabolic labeling studies revealed that FOH and GGOH can become converted to their phosphorylated derivatives (Fpp and GGpp) and utilized for the synthesis of sterols and protein prenylation (**Crick et al., 1997**). Kinases that phosphorylate these isoprenols have been reported in plants and bacteria (**Ohnuma et al., 1996**; **Thai et al., 1999**); however, mammalian counterparts of these enzyme remain to be identified and characterized. In contrast, activities that dephosphorylate isoprenyl pyrophosphates in mammalian cells have been identified. Type 1 polyisoprenoid diphosphate phosphatase (PDP1) is an integral ER membrane protein with a cytosolic active site that preferentially hydrolyzes isoprenyl pyrophosphates such as Fpp and GGpp to generate FOH and GGOH, respectively

(*Fukunaga et al., 2006*; *Miriyala et al., 2010*). Overexpression of PDP1 reduces intracellular levels of isoprenyl pyrophosphates and inhibits protein prenylation. In the current study, we explore a role for PDP1 in the geranylgeranyl-regulated ERAD of reductase and ER-to-Golgi transport of UBIAD1. The results of these studies indicate that PDP1 contributes to a pathway for interconversion of isoprenols and their phosphorylated derivatives that helps to balance sterol and nonsterol branches of the mevalonate pathway.

## Results

### PDP1 overexpression modulates ERAD of HMG CoA reductase and ER-to-Golgi transport of UBIAD1

We began the exploration into a role for PDP1 in sterol-accelerated ERAD of reductase by generating a line of cells designated SV-589/PDP1-Myc-FLAG that stably express tetracycline-inducible PDP1 tagged at the C-terminus with Myc and FLAG epitopes. In *Figure 1A*, SV-589/PDP1-Myc-FLAG cells were metabolically labeled with azido-GGOH, which is converted into a pyrophosphate derivative that can be utilized for protein geranylgeranylation (*Chan et al., 2009*; *Palsuledesai et al., 2016*). The cells were then harvested, lysed, and precipitated, after which azido-geranylgeranylated proteins were labeled with biotin-alkyne through a click reaction. Analysis of samples through streptavidin-HRP blotting revealed the presence of azido-geranygeranylated proteins in the 20–25 kDa range (*Figure 1A*, lanes b and c). Protein azido-geranylgeranylation was abolished by tetracycline-mediated induction of PDP1-Myc-FLAG (lanes e and f), suggesting the modification was inhibited by PDP1-catalyzed hydrolysis of azido-geranylgeranyl pyrophosphate as previously reported (*Miriyala et al., 2010*).

In *Figure 1B*, SV-589/PDP1-Myc-FLAG cells were depleted of nonsterol isoprenoids through incubation in sterol-replete medium containing fetal calf serum (FCS) and the reductase inhibitor compactin. Some of the cells were also treated with tetracycline to induce expression of PDP1-Myc-FLAG. After 16 hr, cells were treated with various concentrations of GGOH and subsequently harvested for subcellular fractionation and immunoblot analysis. The results show that GGOH enhanced ERAD of reductase, which was indicated by its disappearance from membranes (*Figure 1B*, compare lane a with lanes b-e). This GGOH-enhanced ERAD of reductase was blunted when overexpression of PDP1-Myc-FLAG was induced by tetracycline (compare lane f with lanes g-j). Serine-212 of PDP1 lies within the phosphatase catalytic motif; its mutation to threonine (S212T) abolishes enzymatic activity (*Miriyala et al., 2010*). In the experiment shown in *Figure 1—figure supplement 1*, SV-589/TR cells that stably express the tetracycline repressor (TR) were transfected with an expression plasmid encoding the membrane domain of reductase, which is both necessary and sufficient for ERAD (*Sever et al., 2003b*), and cultured in FCS. Following transfection, cells were treated with tetracycline in the absence or presence of GGOH. As expected, GGOH enhanced the ERAD of the membrane domain of reductase (*Figure 1—figure supplement 1*, lanes 1 and 2); this ERAD was enhanced upon co-expression of catalytically inactive PDP1 (S212T)-FLAG (lanes 3 and 4). *Figure 1C* shows results of an experiment in which SV-589/PDP1-Myc-FLAG cells were first depleted of both sterol and nonsterol isoprenoids through incubation in medium containing lipoprotein deficient serum (LPDS) and compactin. Subsequent treatment of the cells with the oxysterol 25-hydroxycholesterol (25-HC) stimulated reductase ERAD as expected (*Figure 1C*, compare lanes a and b). The oxysterol also blocked proteolytic activation of sterol regulatory binding-protein-2 (SREBP-2), which was indicated by the disappearance of proteolytically released fragments of SREBP-2 from nuclei and concurrent accumulation of its membrane-bound precursor (compare lanes a and b). Consistent with our previous observations (*Sever et al., 2003a*), GGOH enhanced ERAD of reductase that was stimulated by 25-HC (lanes d and e). The oxysterol continued to accelerate reductase ERAD upon inducible overexpression of PDP1-Myc-FLAG; however, the effect of GGOH on this ERAD was diminished (compare lanes b-e with g-j).

We next used immunofluorescence microscopy to assess the subcellular localization of endogenous UBIAD1 in SV-589/PDP1-Myc-FLAG cells. As we previously established in several studies (*Schumacher et al., 2015*; *Schumacher et al., 2016*; *Schumacher et al., 2018*), UBIAD1 exhibited a reticular localization indicative of the ER when cells were depleted of nonsterol isoprenoids through incubation in compactin (*Figure 2A*, panel 1). The addition to the cells of GGOH triggered the translocation of UBIAD1 from the ER to the Golgi as expected (panels 2–4). GGOH-induced translocation of UBIAD1 from ER to Golgi was impeded by tetracycline (compare panels 1–4 with 5–8), which induced

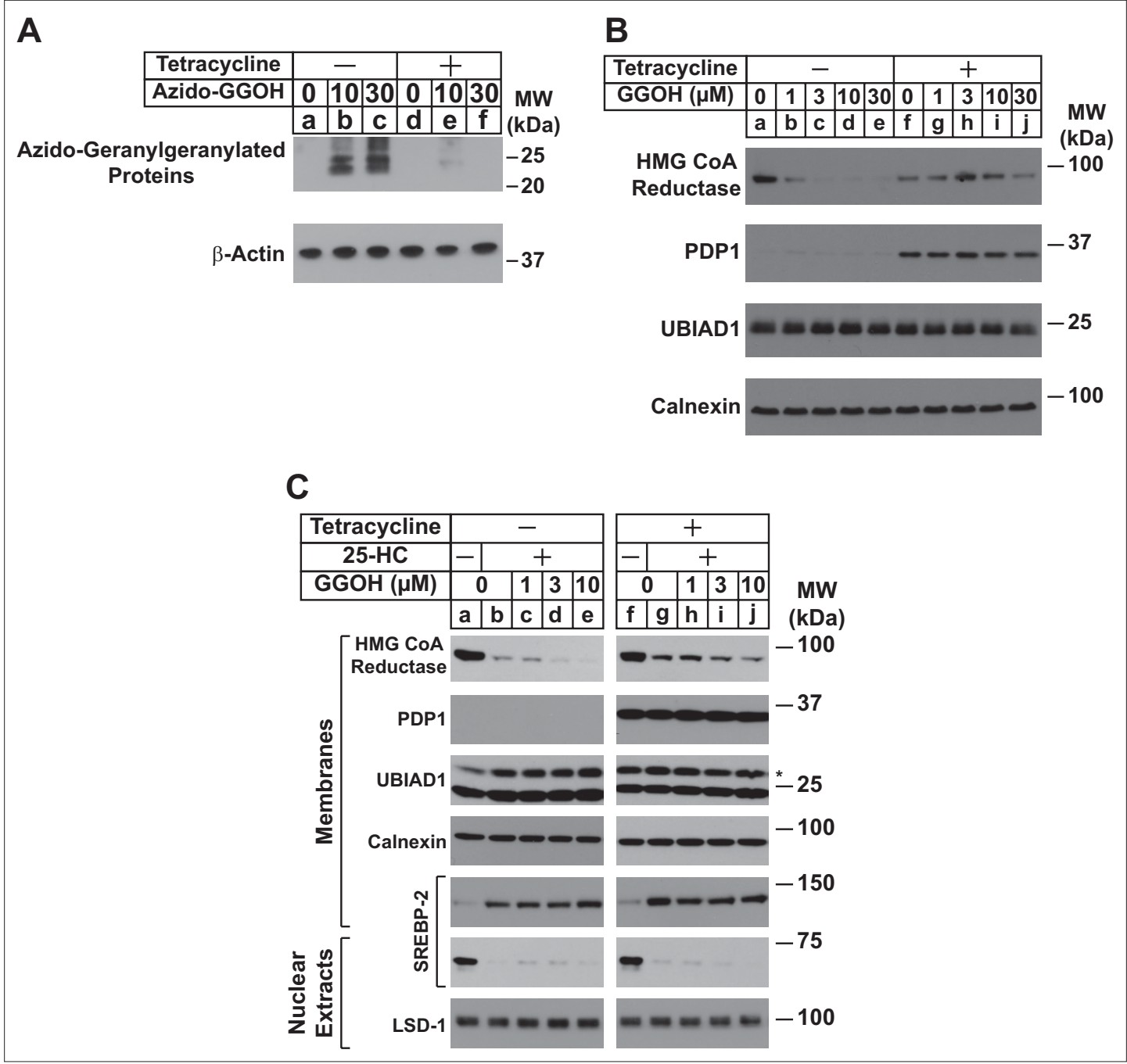

**Figure 1.** Overexpression of PDP1 inhibits protein geranylgeranylation and GGOH-enhanced ERAD of HMG CoA reductase. (**A**) SV-589/PDP1-Myc-FLAG cells were set up on day 0 at $2 \times 10^5$ cells per 60 mm dish in medium A supplemented with 5 % FCS. On day 1, cells were refed identical medium in the absence or presence of tetracycline (1 µg/ml) and the indicated amount of azido-geranylgeraniol (azido-GGOH). Following incubation for 16 hr at 37 °C, cells were harvested, lysed and proteins were precipitated. The resulting material was then resuspended in buffer, labeled with biotin alkyne, and subjected to SDS-PAGE followed by streptavidin blotting as described in 'Materials and methods'. (**B and C**) SV-589/PDP1-Myc-FLAG cells were set up on day 0 at $3 \times 10^5$ cells per 100 mm dish in medium A supplemented with 5 % FCS. (**B**) On day 3, cells were refed the identical medium containing 10 µM sodium compactin and 50 µM sodium mevalonate in the absence or presence of tetracycline (1 µg/ml). Following incubation for 16 hr at 37 °C, cells were treated with the indicated amount of GGOH and incubated an additional 6 hrs at 37 °C. Cells were then harvested, lysed, and subjected to subcellular fractionation. Aliquots of resulting membrane fractions (50 µg protein loaded/lane) were subjected to SDS-PAGE, followed by immunoblot analysis with IgG-A9 (against reductase), anti-PDP1 IgG, IgG-1H12 (against UBIAD1), and anti-calnexin IgG. (**C**) On day 3, cells were depleted of sterol and nonsterol isoprenoids through incubation in medium A containing 10 % LPDS, 10 µM compactin, and 50 µM mevalonate; some of the cells also received 1 µg/ml tetracycline as indicated. After 16 hr at 37 °C, the cells were treated with or without 1 µg/ml 25-HC and the indicated amount of

*Figure 1 continued on next page*

*Figure 1 continued*

GGOH. Following incubation for 3 hr at 37 °C, cells were harvested, lysed, and subjected to subcellular fractionation. Aliquots of resulting membrane and nuclear extract fractions (30–50 µg protein loaded/lane) were analyzed by immunoblot using IgG-A9 (against reductase), anti-PDP1 IgG, IgG-1H12 (against UBIAD1), anti-calnexin IgG, IgG-22D5 (against SREBP-2), and anti-LSD-1 IgG.

The online version of this article includes the following figure supplement(s) for figure 1:

**Figure supplement 1.** Overexpression of catalytically inactive PDP1 (S212T) enhances ERAD of HMG CoA reductase.

overexpression of PDP1-Myc-FLAG (compare panels 9–12 with 13–16). *Figure 2—figure supplement 1* shows an experiment in which SV-589/TR cells were transfected with UBIAD1-Myc in the absence or presence of PDP1 (S212T)-FLAG and cultured in FCS-containing medium supplemented with tetracycline. In mock-transfected cells, UBIAD1-Myc localized to the Golgi of the isoprenoid-replete cells (*Figure 2—figure supplement 1*, panel 1) and compactin caused the protein's relocalization to the ER (panel 2). In the absence of compactin, UBIAD1-Myc localized to the Golgi upon co-expression with PDP1 (S212T)-Myc (panel 3); however, the protein remained in the Golgi upon compactin-mediated depletion of nonsterol isoprenoids (panel 4).

Results of *Figures 1 and 2A* indicate that overexpression of PDP1 accelerates the hydrolysis of GGpp and thereby blunts ERAD of reductase and ER-to-Golgi transport of UBIAD1 stimulated by the addition to cells of GGOH. Structural studies indicate some SCD-associated mutations in UBIAD1 reduce the enzyme's affinity for GGpp (*Cheng and Li, 2014*; *Huang et al., 2014*). Thus, we next used a previously described in vitro assay (*Schumacher et al., 2016*) to compare GGpp-induced incorporation of wild type and SCD-associated UBIAD1 into ER-derived transport vesicles. In *Figure 2B*, UBIAD1-deficient cells were transfected with expression plasmids encoding either Myc-tagged, wild-type UBIAD1 or the SCD-associated N102S variant of the enzyme. Following transfection, the cells were deprived of sterol and nonsterol isoprenoids to trap UBIAD1 in the ER and subsequently harvested for subcellular fractionation. Isolated membrane fractions were then incubated in vitro in the absence or presence of exogenous cytosol, which provided COPII proteins required for ER-to-Golgi transport of proteins (*Rexach and Schekman, 1991*; *Rowe et al., 1996*), an ATP/GTP regeneration system, and varying amounts of GGpp. Following the in vitro reaction, donor membranes and budded vesicles were isolated for immunoblot analysis. The results show that GGpp caused wild-type UBIAD1 to become incorporated into vesicles (*Figure 2B*, lanes a-e); however, UBIAD1 (N102S) was refractory to the GGpp-induced reaction (lanes f-j). We then measured GGpp-induced incorporation of the remaining 19 SCD-associated UBIAD1 variants into ER-derived transport vesicles. The results show that again, GGpp triggered incorporation of wild-type UBIAD1 into vesicles (*Figure 2C*, lanes 2 and 14), whereas the efficiency of GGpp-induced vesicle incorporation was reduced for all 20 SCD-associated variants of the prenyltransferase (lanes 3–12; 15–24).

## PDP1 knockdown modulates ERAD of HMG CoA reductase and ER-to-Golgi transport of UBIAD1

RNA interference (RNAi) was next used to examine a role for endogenous PDP1 in regulation of GGpp pools that control the ERAD of reductase and ER-to-Golgi transport of UBIAD1. In *Figure 3A*, SV-589 cells cultured in FCS-containing medium were transfected with small interfering RNAs (siRNAs) designated A-E targeting PDP1 or green fluorescent protein (GFP), which is not expressed in the cells. Immunoblot analysis of detergent lysates prepared from the transfected cells revealed efficient knockdown of PDP1 protein by five different siRNAs (*Figure 3A*, compare lanes 1 with 2–6). PDP1 knockdown caused reductase to disappear from membranes (lanes 2–6), indicating the protein's enhanced ERAD. *Figure 3B* compares the ability of GGOH to augment reductase ERAD in control and PDP1-knockdown cells cultured in FCS. RNAi-mediated knockdown of PDP1 enhanced the ERAD of reductase from membranes as expected (*Figure 3B*, compare lanes a and b). Chronically depleting the cells of nonsterol isoprenoids through incubation with compactin caused reductase to accumulate in both control-transfected cells (compare lanes a and c) and PDP1 knockdown cells (compare lanes b and g). The compactin-mediated accumulation of reductase was reversed by the addition of GGOH to control-transfected cells (lanes d-f) and this reversal was augmented by the knockdown of PDP1 (compare lane c-f with lanes g-j). Similar results were obtained in the cycloheximide-chase experiment shown in *Figure 3—figure supplement 1A* (compare lanes c-f with g-j). Moreover, the

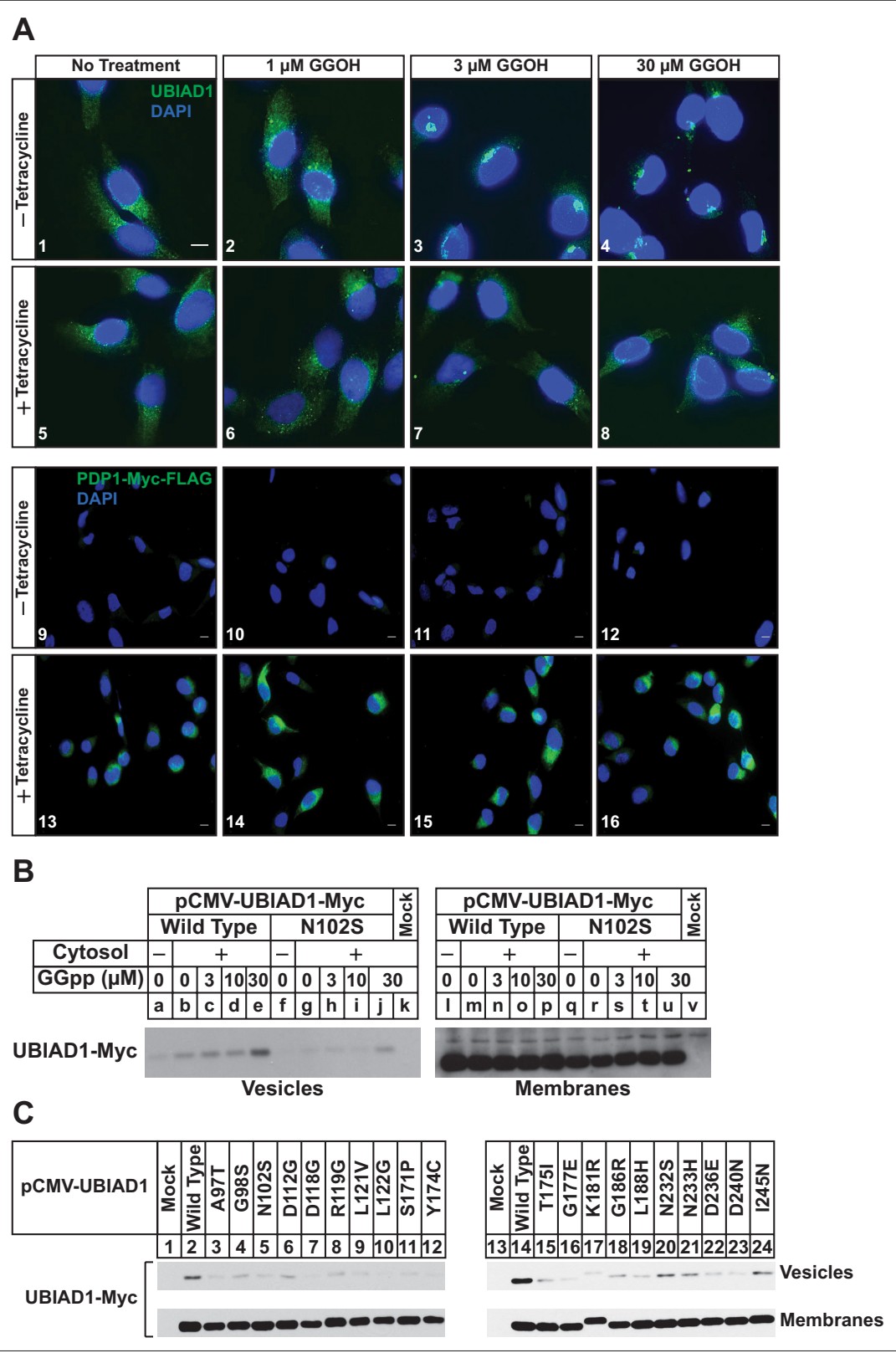

**Figure 2.** PDP1 overexpression blunts GGOH-induced, ER-to-Golgi transport of UBIAD1. (**A**) SV-589/PDP1-Myc-FLAG cells were set up on day 0 at 8 × 10⁴ cells per well of 6-well plates with coverslips in medium A supplemented with 5 % FCS. On day 1, cells were switched to medium A containing 5 % FCS, 10 μM compactin, 50 μM mevalonate in the absence or presence of 1 μg/ml tetracycline. After 16 hr at 37 °C, the cells were

*Figure 2 continued on next page*

*Figure 2 continued*

treated with the indicated amount of GGOH for an additional 6 hr, after which they were fixed and analyzed by immunofluorescence microscopy using IgG-1H12 (against UBIAD1) using a Zeiss Axio Observer Epifluorescence microscope as described in 'Materials and methods'. Cells were also stained with DAPI to visualize nuclei. Lower magnification images are shown for anti-Myc-probed samples to illustrate tetracycline-induced expression of PDP1-Myc-FLAG. (**B and C**) SV-589 (ΔUBIAD1) cells were set up on day 0 at $6 \times 10^5$ cells per 100 mm dish in medium A supplemented with 5 % FCS. On day 2, cells were transfected with pCMV-Myc-UBIAD1 (2 µg/dish wild type or 0.5 µg/dish indicated mutant) as described in 'Materials and methods'. Four hours after transfection, the cells received a direct addition of medium A containing 5 % LPDS, 10 µM compactin, and 50 µM mevalonate (final concentrations) and incubated for an additional 16 hr at 37 °C. The cells were subsequently harvested for preparation of microsomes that were incubated at 37 °C in buffer containing ATP, GTP, and an ATP-regenerating system in absence or presence of UT-2 cytosol and the indicated concentration (**B**) or 30 µM GGpp (**C**). After 20 min, reactions were centrifuged to obtain vesicles and membranes, which were subjected to SDS-PAGE followed by immunoblot analysis with IgG-9E10 (against Myc-UBIAD1).

The online version of this article includes the following figure supplement(s) for figure 2:

**Figure supplement 1.** Overexpression of PDP1 (S212T) blocks compactin-induced relocation of UBIAD1 from Golgi to ER.

GGOH-enhanced ERAD of reductase was blunted with PDP1 knockdown cells were treated with the proteasome inhibitor MG-132 (*Figure 3—figure supplement 1B*, compare lanes a-c with e-h). It should be noted that PDP1 siRNA-E was used for experiments in the remainder of this study.

Our previous studies showed that treatment of isoprenoid-replete cells with compactin caused the redistribution of UBIAD1 from the Golgi to ER (*Schumacher et al., 2016*). Moreover, a similar redistribution of UBIAD1 to the ER was observed when export from the ER was inhibited in isoprenoid-replete cells. These observations led us to conclude that UBIAD1 shuttles between the ER and Golgi of cells and becomes trapped in the ER upon sensing depletion of GGpp in membranes of the organelle (*Figure 4A*). Retrograde transport of UBIAD1 from the Golgi to ER is evident as early as 1 hr following compactin treatment, indicating that ER-localized pools of GGpp that govern transport of UBIAD1 to the Golgi turnover rapidly. Thus, we next designed a set of experiments to evaluate the effect of PDP1 knockdown on the rapid, compactin-induced redistribution of UBIAD1 from the Golgi to ER and inhibition of reductase ERAD. *Figure 4B* shows that when cells were transfected with GFP siRNA and cultured in sterol-replete FCS-containing medium, a 2 hr compactin treatment caused UBIAD1 to redistribute from the Golgi to the ER as previously reported (panels 1 and 3). In parallel studies, compactin treatment also caused reductase to become stabilized in cells transfected with GFP siRNA (*Figure 4C*, compare lane a with lanes b-d). UBIAD1 continued to localize to the Golgi of untreated PDP1 knockdown cells (*Figure 4B*, panel 2). However, the knockdown of PDP1 prevented both the compactin-induced redistribution of UBIAD1 from the Golgi to ER (*Figure 4B*, compare panels 2 and 4) and stabilization of reductase (*Figure 4C*, compare lane e with lanes f-h). *Figure 4D* shows that when cells were depleted of both sterol and nonsterol isoprenoids through incubation in medium supplemented with LPDS plus compactin, reductase became stabilized in control- and PDP-1 knockdown cells (compare lanes a and e with lanes b-d and f-h, respectively). When cells were cultured in LPDS plus 25-HC (to satisfy cellular requirements for sterols), PDP1 knockdown abolished compactin-induced stabilization of reductase (*Figure 4E*, compare lanes a-c with d-f).

## PDP1 hydrolyzes pools of GGpp required for protein geranylgeranylation

To appraise the significance of PDP1 in regulation of total intracellular pools of GGpp, we measured levels of the geranylgeranylated small GTPases RhoA, Rap1, and Cdc42 as well as reductase following incubation of cells in the absence or presence of compactin and/or GGOH. The experiment of *Figure 5A* shows that compactin caused the marked accumulation of RhoA and Cdc42 (compare lane 1 with lanes 6 and 8). Compactin also caused the appearance of a slower migrating, unprenylated form of Rap1 (lanes 6 and 8). The effect of compactin on the accumulation of RhoA and Cdc42 and migration of Rap1 was reversed by the addition to cells of GGOH (*Figure 5A*, compare lanes 6 and 8 with lanes 7 and 9). GGOH also prevented compactin-induced stabilization of reductase (compare lanes 2, 4, 6, and 8 with lanes 3, 5, 7, and 9). Importantly, FOH failed to prevent compactin-induced

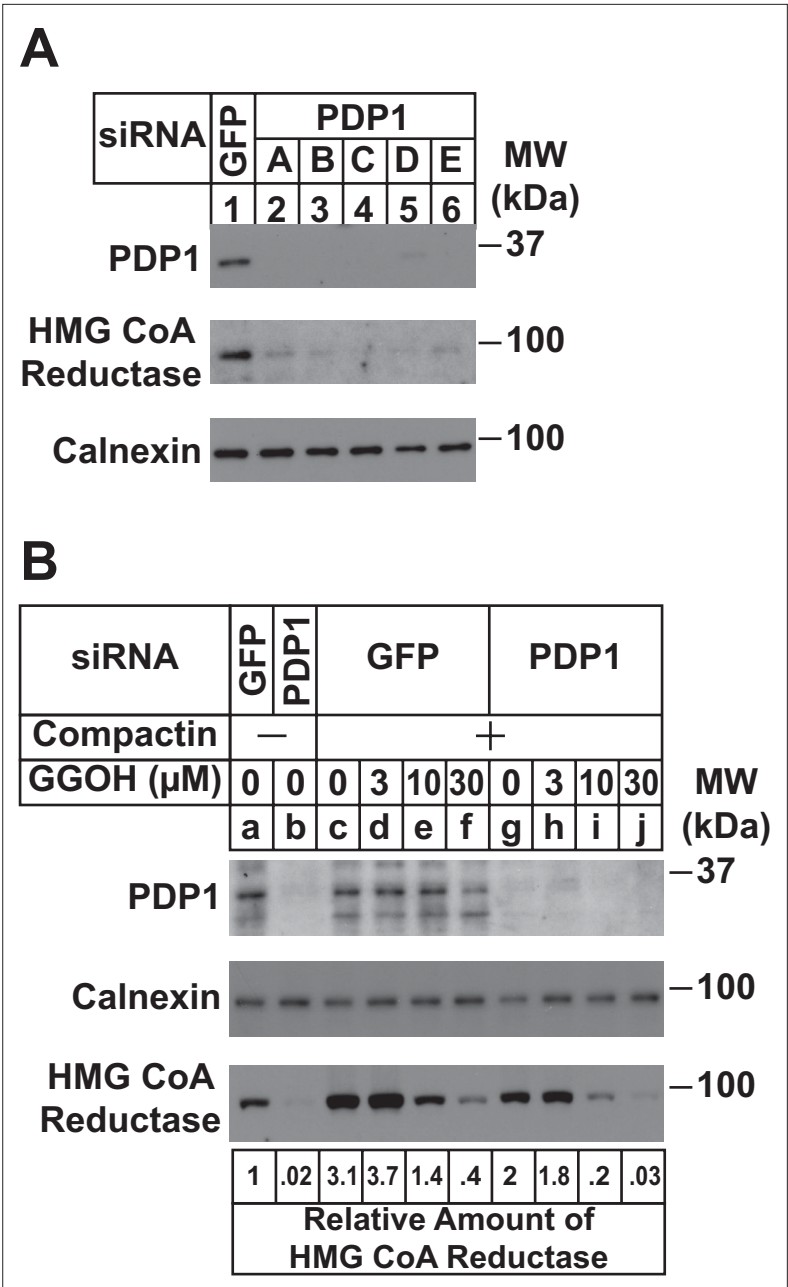

**Figure 3.** RNAi-mediated knockdown of PDP1 enhances GGpp-induced ERAD of HMG CoA reductase. SV-589 cells were set up on day 0 at a density of 1.5 × 105 cells per 60 mm dish in medium A supplemented with 5 % FCS. On day 1 cells were transfected in medium A containing 5 % FCS with siRNAs targeting mRNAs that encode GFP or PDP1. In (**A**), cells were transfected with PDP1 siRNAs **A–E**. (**A**) On day 3, cells were harvested, lysed, and aliquots of resulting whole-cell lysates (10 µg protein loaded/lane), followed by immunoblot analysis using anti-PDP1, anti-calnexin, and IgG-A9 (against reductase). (**B**) On day 3, cells were switched to medium A supplemented with 5 % FCS in the absence or presence of 10 µM compactin. Following incubation for 24 hr at 37 °C, cells were refed the identical medium and the indicated concentration of GGOH and incubated for an additional 24 hr. Cells were then harvested for subcellular fractionation; aliquots of resulting membrane fractions (15 µg protein loaded/lane) were subjected to SDS-PAGE and immunoblot analysis as described in (**A**). The amount of reductase was determined by quantifying the band corresponding to reductase using ImageJ software. Each value represents the amount of reductase protein relative to that in untreated cells transfected with GFP siRNA, which was arbitrarily set as 1.

The online version of this article includes the following figure supplement(s) for figure 3:

**Figure supplement 1.** Knockdown of PDP1 enhances proteasome-mediated ERAD of HMG CoA reductase.

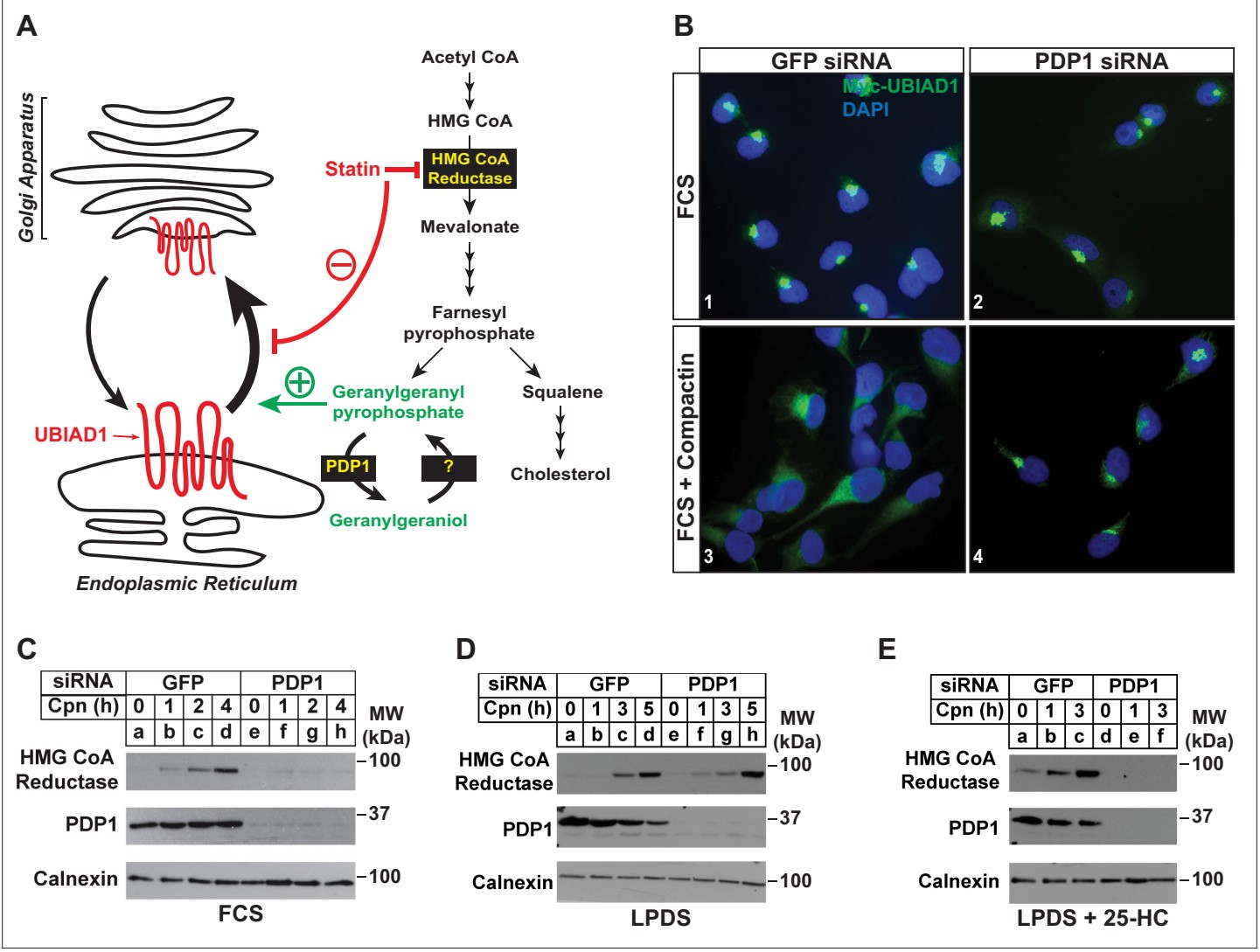

**Figure 4.** RNAi-mediated knockdown of PDP1 abolishes compactin-induced, Golgi-to-ER redistribution of UBIAD1 and stabilization of HMG CoA reductase. (**A**) Proposed role of PDP1 in modulating levels of GGpp that stimulate transport of UBIAD1 from membranes of the ER to Golgi. (**B**) SV-589/pMyc-UBIAD1 cells were set up on day 0 at $7.5 \times 10^4$ cells per well of 6-well plates with glass coverslips in medium A supplemented with 5 % FCS. On day 1, the cells were transfected in identical medium with siRNAs against GFP or PDP1 mRNAs as described in the legend to *Figure 3*. Following incubation for 16 h at 37 °C, cells were switched to medium A supplemented with 5 % FCS in the absence or presence 10 µM compactin. After 2 hr, cells were fixed, permeabilized, and analyzed by immunofluorescence microscopy using IgG-9E10 (against Myc-UBIAD1) as described in the legend to *Figure 2*. (**C–E**) SV-589 cells were set up on day 0 at $2 \times 10^5$ cells per 60 mm dish in medium A containing 5 % FCS. On day 1, cells were transfected in identical medium with siRNAs against GFP or PDP1. On day 2, cells were refed medium A supplemented with either 5 % FCS (**C**) or 10 % LPDS (**D and E**). Following incubation for 16 hr at 37 °C, the cells were treated with 10 µM compactin in the absence (**C and D**) or presence (**E**) of 1 µg/ml 25-HC. The cells were then incubated for the indicated period of time, after which detergent lysates were prepared and subjected to immunoblot analysis using IgG-A9 (against reductase), anti-PDP1, and anti-calnexin.

accumulation of reductase, RhoA, and Cdc42 (*Figure 5B*, compare lanes 2 and 6). Inhibition of proteasomal degradation with MG-132 blocked the GGOH-dependent reversal of compactin-induced stabilization of reductase, RhoA, and Cdc42 (*Figure 5C*, compare lanes 2 and 4 with 6 and 8; *Figure 5D*, compare lanes 2 and 4 with 5 and 6), which is consistent with previous reports (*Holstein et al., 2002*; *Stubbs and Von Zee, 2012*; *Von Zee et al., 2009*). RhoA and Cdc42 were also stabilized when cells were treated with the geranylgeranyl transferase type 1 inhibitor, GGTI (*Figure 5D*, lanes 7 and 8).

Using RhoA stability as an indicator of intracellular levels of GGpp, we next subjected control and PDP1 knockdown cells to treatment with compactin followed by immunoblot analysis (*Figure 6A*). The results revealed the expected stabilization of RhoA and reductase in control-transfected cells

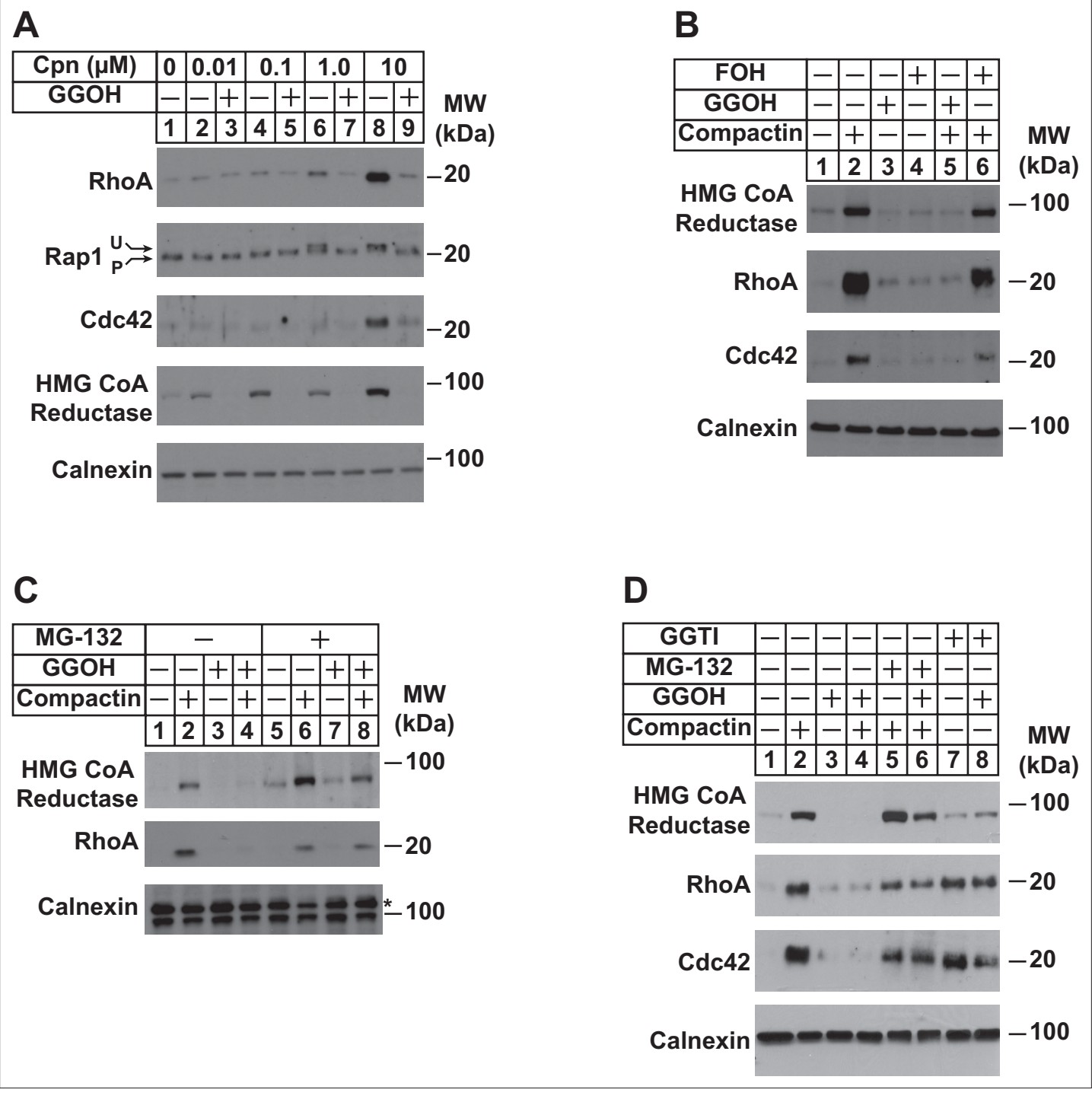

**Figure 5.** Depletion of GGpp and subsequent inhibition of geranylgeranylation blocks proteasomal degradation of the small GTPases RhoA and cdc42. SV-589 cells were set up on day 0 at a density of 1.5 × 10^5 cells per 60 mm dish in medium A supplemented with 5 % FCS. On day 2, cells were washed and refed identical medium supplemented with 0–10 μM (**A**) or 10 μM compactin (**B–D**). Following incubation for 24 hr at 37 °C, the cells received the identical medium in the absence or presence of 20 μM GGOH, 20 μM FOH, 10 μM MG-132, and 10 μM GGTI as indicated and incubated an addition 24 hr. Cells were then harvested for preparation of detergent-solubilized lysates that were subjected to SDS-PAGE, followed by immunoblot analysis with anti-RhoA, anti-cdc42, anti-Rap1, IgG-A9 (against reductase), and anti-calnexin. Prenylated (**P**) and unprenylated (**U**) forms of Rap1 protein are denoted by arrows.

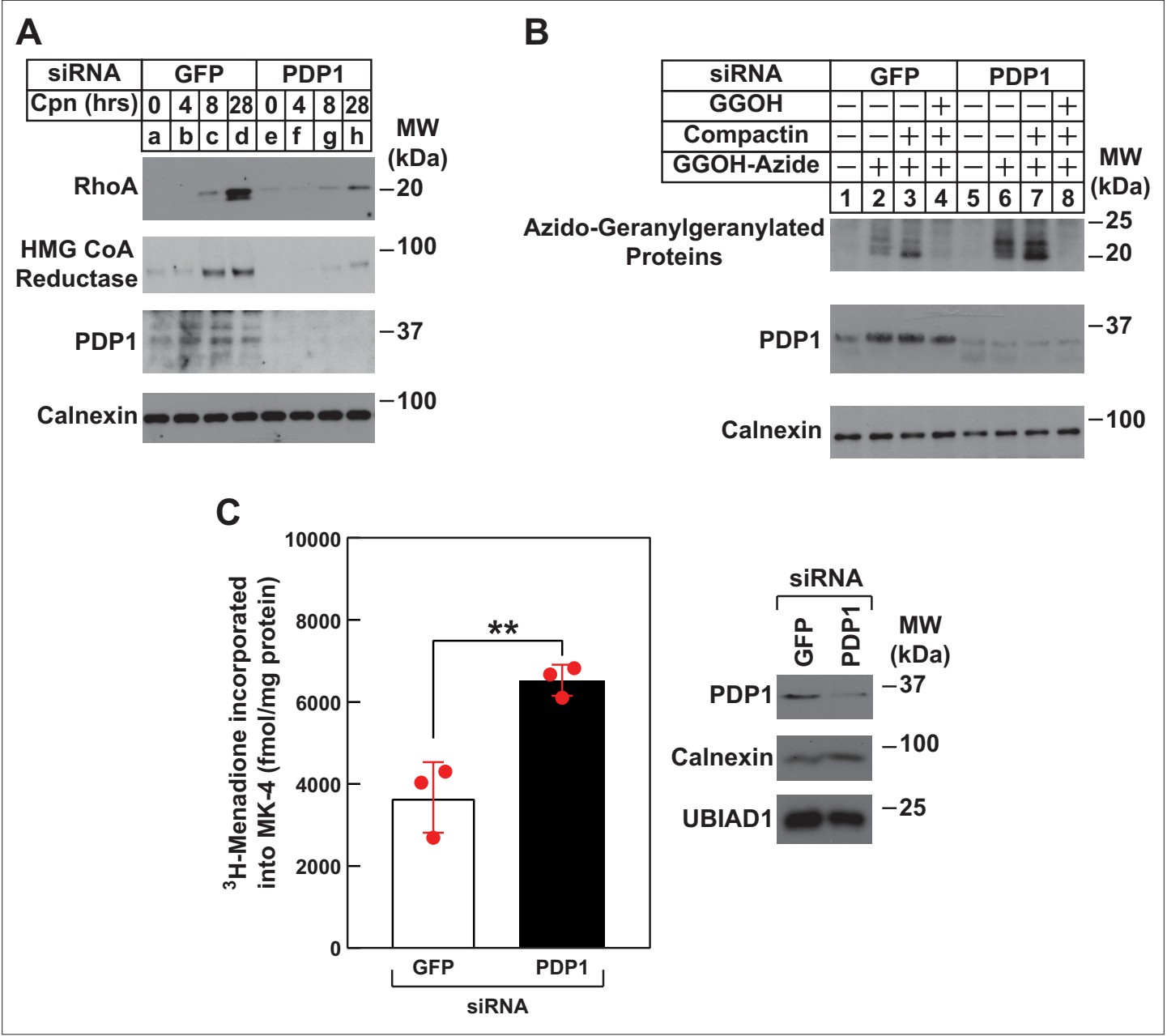

**Figure 6.** RNAi-mediated knockdown of PDP1 blunts compactin-induced stabilization of RhoA, enhances protein geranylgeranylation, and augments synthesis of menaquinone-4 (MK-4). (**A and B**) SV-589 cells were set up on day 0 at 1.5 × 10⁵ cells per 60 mm dish in medium A supplemented with 5 % FCS. On day 1, cells were transfected in the identical medium with siRNAs against mRNAs encoding GFP and PDP1 as described in the legend to *Figure 3*. (**A**) On day 2, transfected cells were washed and refed medium A containing 5 % FCS and 10 µM compactin, after which they were incubated for the period of indicated time at 37 °C. Following incubations, cells were harvested for preparation of detergent-solubilized lysates that were subjected to immunoblot analysis with anti-RhoA, IgG-A9 (against reductase) and anti PDP1, and anti-calnexin (1:5000). (**B**) On day 2, transfected cells were refed medium A supplemented with 5 % FCS in the absence or presence of 30 µM azido-GGOH, 10 µM compactin, and 15 µM GGOH as indicated. Following incubation for 24 hr at 37 °C, cells were harvested, lysed and proteins were precipitated. The resulting material was resuspended in buffer, labeled with biotin alkyne as described in the legend to *Figure 1*. The samples were then subjected to SDS-PAGE followed by streptavidin blotting or immunoblot analysis with anti-PDP1 and anti-calnexin. (**C**) SV589 cells were set up on day 0 at 2.5 × 10⁵ cells per 60 mm dish in medium A containing 5 % FCS. On day 1, cells were transfected with siRNAs targeting mRNAs encoding GFP or PDP1 as indicated and described in 'Materials and methods'. On day 2, the cells were treated with 10 nM [³H]-menadione for 12 hours. Following the incubation, the cells were harvested, lysed, and lipids were extracted as described in 'Materials and methods'. The amount of [³H]menadione-incorporated into MK-4 was determined by TLC and scintillation counting. The values shown are the mean of triplicate samples (standard error). Aliquots of whole cell lysate (20 µg protein/lane) were subjected to SDS-PAGE, and immunoblot analysis was carried with IgG-PDP1.

treated with compactin (**Figure 6A**, compare lanes a and d). However, the compactin-induced stabilization of RhoA and reductase was noticeably blunted in PDP1 knockdown cells (compare lanes a and d with e and h). In **Figure 6B**, control and PDP1 knockdown cells were metabolically labeled for 24 hr with azido-GGOH in the absence or presence of compactin and exogenous GGOH. Cells were subsequently harvested, lysed, and precipitated after which, azido-geranylgeranylated proteins were labeled with biotin-alkyne. Streptavidin-HRP blotting of resulting material revealed the presence of azido-geranylgeranylated proteins (molecular weight, 20–25 kDa) in reactions from GFP siRNA-transfected cells (**Figure 6B**, lane 2). Azido-geranylgeranylation of these proteins was enhanced when cells were treated with compactin (lane 3) and eliminated by inclusion of GGOH in labeling medium (lane 4). The level of azido-geranylgeranylated proteins was noticeably enhanced in PDP1 knockdown cells (compare lanes 2 and 6). The modification was further enhanced by compactin (lane 7) and eliminated by exogenous GGOH (lane 8). Finally, we used synthesis of MK-4 as another surrogate for production of GGpp in PDP1 knockdown cells. **Figure 6C** shows an experiment in which control and PDP1 knockdown cells were metabolically labeled with [$^3$H]menadione, after which the cells were lysed and incorporation of [$^3$H]menadione into MK-4 was determined. The results show that PDP1 knockdown led to the elevated production of [$^3$H]MK-4 (1.8-fold) compared to that in controls, which likely results from enhanced intracellular accumulation of GGpp.

## Discussion

We previously speculated that the effect of GGOH on ERAD of reductase and ER-to-Golgi transport of UBIAD1 was indirect and required its conversion to GGpp by an unknown kinase(s). Stimulated by results from Morris and co-workers that showed the overexpression of PDP1 led to depletion of GGpp (and likely other polyisoprenyl pyrophosphates) (**Miriyala et al., 2010**), we designed a set of experiments to assess a role for PDP1 in GGOH-mediated regulation of reductase and UBIAD1. The results of these studies reveal that indeed, overexpression of PDP1 abrogated both the ERAD of reductase and translocation of UBIAD1 from the ER to Golgi that was stimulated by the addition to cells of GGOH (**Figures 1B, C and 2A**). Overexpression of catalytically inactive PDP1 (S212T) accelerated ERAD of reductase (**Figure 1—figure supplement 1**) and prevented the statin-induced translocation of UBIAD1 from the Golgi to ER (**Figure 2—figure supplement 1**). This indicates that PDP1 (S212T) blocks the activity of endogenous PDP1 in a dominant-negative fashion, causing GGpp to accumulate. These findings infer that overexpressed PDP1 hydrolyzes pools of GGpp required for protein geranylgeranylation (**Figure 1A**) and membrane-embedded pools of the isoprenyl pyrophosphate that promote ER-to-Golgi transport of UBIAD1 and stimulate the ERAD of reductase. This notion is further supported by results showing that SCD-associated UBIAD1 resisted to GGpp-induced incorporation into ER-derived transport vesicles (**Figure 2B and C**). Notably, the structural analysis of archaeal UbiA prenyltransferases indicate that SCD-associated mutations in UBIAD1 (such as N102S) surround the enzyme's membrane-encased active site and some directly interact with the pyrophosphate group of GGpp (**Cheng and Li, 2014**; **Huang et al., 2014**). Thus, it would be expected that many SCD-associated mutations disrupt binding of UBIAD1 to GGpp, explaining ER sequestration and reduced MK-4 synthetic activity of SCD-associated variants (**Jun et al., 2020**).

We found previously that treatment of isoprenoid-replete cells with compactin for as little as 1 hr caused UBIAD1 to redistribute from the Golgi to ER (**Schumacher et al., 2016**, see **Figure 4A**), indicating pools of GGpp that are embedded within membranes of the ER turnover rapidly. The first line of evidence that PDP1 mediates this rapid turnover is provided by **Figure 3A**, which shows that the RNAi-mediated knockdown of PDP1 augmented ERAD of reductase in cells replete with sterols. PDP1 knockdown also enhanced GGOH-induced ERAD of reductase in sterol-replete cells treated with compactin (**Figure 3B**). **Figure 4** provides additional evidence for a prominent role of PDP1 in governing levels of ER membrane-embedded GGpp. These studies not only confirmed that compactin caused the redistribution of UBIAD1 from the Golgi to the ER, but they also revealed that this redistribution (**Figure 4B**) as well as the compactin-induced stabilization of reductase (**Figure 4C**) were abolished by knockdown of PDP1. Together, these results indicate that levels of GGpp embedded within ER membranes are maintained by the combined actions of ER-to-Golgi transport of UBIAD1, ERAD of reductase, and PDP1-mediated hydrolysis.

Our studies also provide evidence that PDP1 not only hydrolyzes regulatory pools of GGpp within ER membranes, but it also modulates total pools of the isoprenyl pyrophosphate. Some

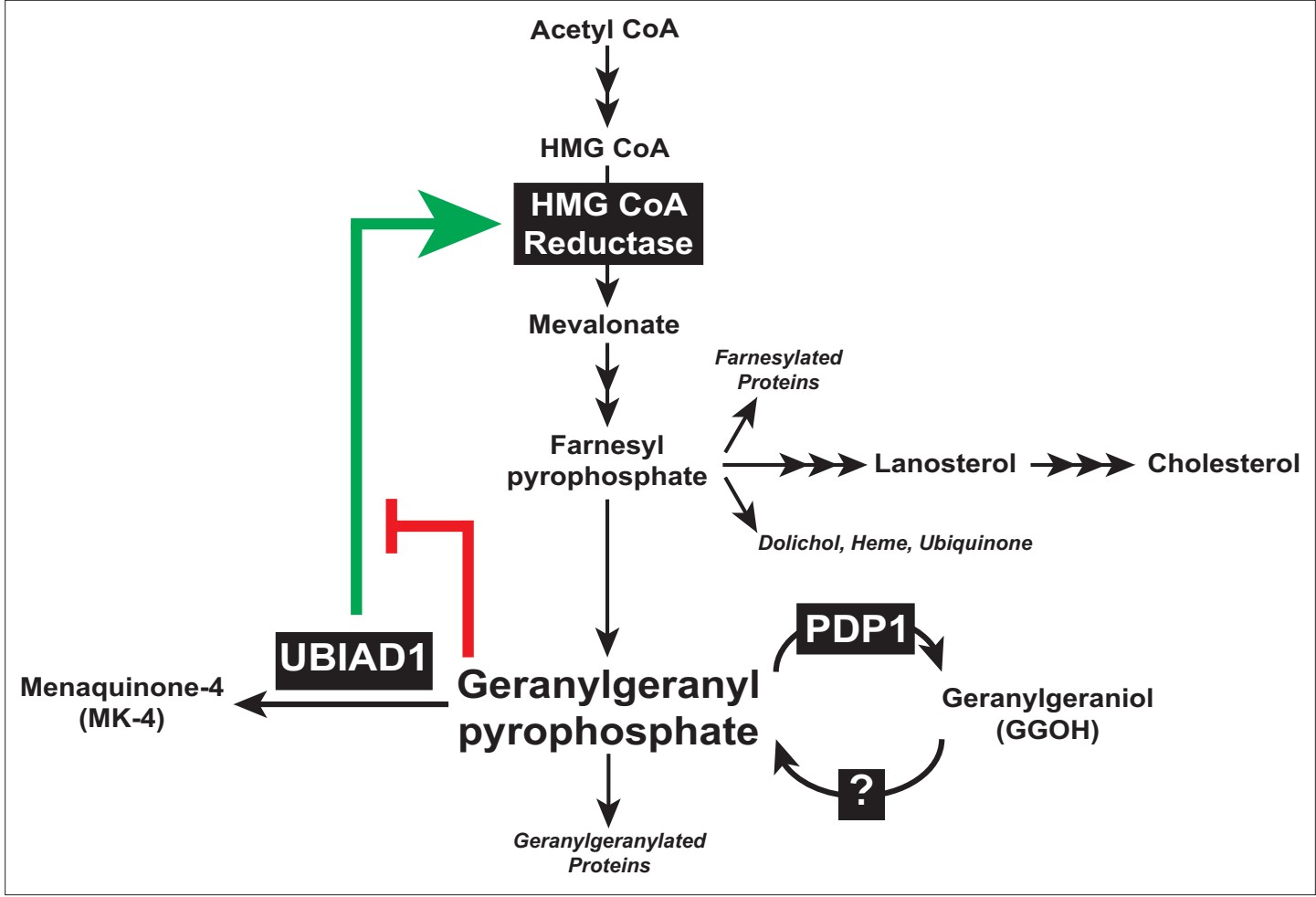

**Figure 7.** Contribution of PDP1 and UBIAD1 to the feedback regulatory system that targets HMG CoA reductase ERAD to control synthesis of GGpp. The synthesis and hydrolysis of GGpp are key focal points in the feedback regulation of the nonsterol branch of the mevalonate pathway (see text for details).

geranylgeranylated small GTPases such as RhoA and Cdc42 become stabilized in the presence of compactin, owing to depletion of GGpp and reduced geranylgeranylation (*Figure 5*). This statin-induced stabilization was blunted in PDP1 knockdown cells (*Figure 6A*), indicating the phosphatase mediates turnover of pools of GGpp that are required for protein geranylgeranylation. This conclusion is further supported by studies that employed azido-GGOH labeling, which shows the total level of protein geranylgeranylation was enhanced in PDP1 knockdown cells (*Figure 6B*). Although not examined in this study, PDP1 likely plays a key role in the interconversion of other isoprenols and isoprenyl pyrophosphates such as FOH and Fpp. A final piece of evidence that PDP1 modulates interconversion of total pools of GGOH and GGpp is provided by *Figure 6C*, which shows that PDP1 knockdown enhanced the UBIAD1-mediated, GGpp-dependent synthesis of MK-4.

Results of the current study considered together with those previously obtained in cultured cells and genetically-manipulated mice (*Jo et al., 2019*; *Jo et al., 2020*), designate the reductase ERAD pathway as a focal point in feedback control of intracellular levels of GGpp and other nonsterol isoprenoids. Sequestration of UBIAD1 in the ER of GGpp-deprived cells leads to the inhibition of reductase ERAD, which causes the protein to accumulate (indicated by green arrow in *Figure 7*). The accumulation of reductase enhances synthesis of mevalonate for replenishment of GGpp and other nonsterol isoprenoids. The restoration of GGpp to optimal levels within ER membranes triggers release of UBIAD1 from reductase; GGpp-induced release inhibits reductase (indicated by red bar in *Figure 7*) by accelerating its ERAD and stimulates translocation of UBIAD1 to the Golgi. This novel feedback regulatory system helps to ensure the constant production GGpp and other essential

nonsterol isoprenoids, while avoiding their overaccumulation. GGpp and Fpp are utilized in the prenylation of RhoA and other small GTPases, which target the enzymes to intracellular membranes. The membrane association of small GTPases is essential for modulation of downstream effectors that control cellular growth and proliferation, cytoskeletal organization, gene expression, energy metabolism, and vesicular transport (*Wang and Casey, 2016*). Thus, dysregulation of isoprenyl pyrophosphate metabolism may be detrimental to cells as indicated by our failure to generate PDP1-deficient cells using CRISPR/Cas9 methods, which should overaccumulate these molecules and accelerate reductase ERAD. The rapid, PDP1-mediated turnover of GGpp (see *Figure 7*) and other isoprenyl pyrophosphates represents an additional protective mechanism that guards against overaccumulation of mevalonate-derived end-products. It will be important in future studies to identify the kinase(s) that phosphorylate isoprenols to isoprenyl pyrophosphates and determine their role in regulation of UBIAD1 and reductase.

Statin-mediated inhibition of reductase is a routinely prescribed approach to lower circulating levels of cholesterol-rich low-density lipoprotein (LDL) and reduce the incidence of cardiovascular disease (*Goldstein and Brown, 2015*; *Stossel, 2008*). However, statins block production of mevalonate-derived products that mediate feedback regulation of reductase, causing its marked accumulation that limits the efficacy of the drugs (*Engelking et al., 2006*; *Goldberg et al., 1990*; *Schonewille et al., 2016*). We showed previously that accumulation of reductase in livers of statin-treated mice corresponded with sequestration of UBIAD1 in the ER (*Jo et al., 2019*), indicating the response results from depletion of GGpp from ER membranes. This implies that drugs mimicking GGpp in stimulating ER-to-Golgi transport of UBIAD1 would relieve inhibition of ERAD and avoid the overaccumulation of reductase that limits statin therapy. *Figure 4* shows that knockdown of PDP1 prevents both the statin-induced redistribution of UBIAD1 from the Golgi to ER and accumulation of reductase. These results denote PDP1-mediated hydrolysis of GGpp as a target for strategies that improve the effectiveness of statins in lowering cholesterol.

# Materials and methods

**Key resources table**

| Reagent type (species) or resource | Designation | Source or reference | Identifiers | Additional information |
|---|---|---|---|---|
| Cell line (human) | SV-589 | PMID:6091915 | RRID: CVCL_RW34 | SV40 transformed human fibroblasts |
| Cell line (human) | SV-589/PDP1-Myc-FLAG | This paper | N/A | SV-589 cells stably expressing tetracycline-inducible human PDP1-Myc-FLAG |
| Antibody | Anti-SREBP-2 (rabbit polyclonal) | PMID:25896350 | IgG-22D5 | (1–5 µg/ml) |
| Antibody | Anti-UBIAD1 (mouse monoclonal) | PMID:29167270 | IgG-1H12 | (1–5 µg/ml) |
| Antibody | Anti-HMGCR (mouse monoclonal) | PMID:22143767 | IgG-A9 | (1–5 µg/ml) |
| Antibody | Anti-PDP1 (goat polyclonal) | Santa Cruz Biotechnology | Cat#SC-163253; RRID:AB_10842717 | (1–5 µg/ml) |
| Antibody | Anti-RhoA (mouse monoclonal) | Santa Cruz Biotechnology | IgG-26C4; Cat#SC-418; RRID:AB_628218 | (1–5 µg/ml) |
| Antibody | Anti-Rap1 (mouse monoclonal) | Santa Cruz Biotechnology | IgG-E6; Cat#SC-398755 | (1–5 µg/ml) |
| Antibody | Anti-Cdc42 (mouse monoclonal) | Santa Cruz Biotechnology | IgG-B8; Cat#SC-8401; RRID:AB_627233 | (1–5 µg/ml) |
| Antibody | Anti-Calnexin (rabbit polyclonal) | Novus Biologicals | Cat#NB100-1965; RRID:AB_10002123 | (1–5 µg/ml) |
| Antibody | Anti-LSD-1 (rabbit polyclonal) | Cell Signaling Technology | Cat#2139; RRID:AB_2070135 | (1–5 µg/ml) |

*Continued on next page*

*Continued*

| Reagent type (species) or resource | Designation | Source or reference | Identifiers | Additional information |
|---|---|---|---|---|
| Antibody | Anti-T7•Tag Antibody (mouse monoclonal) | Sigma Millipore | IgG2b; Cat#69522; RRID:AB_11211744 | (1–5 µg/ml) |
| Antibody | Anti-FLAG M2 (mouse monoclonal) | Sigma-Aldrich | IgG1; Cat#3165; RRID:AB_262044 | (1–5 µg/ml) |
| Transfected construct (human) | pCMV/TO-PDP1-Myc-FLAG | This paper | Cat#V102020 | Human PDP1-Myc inserted into pCDNA4/TO vector (Thermo Fisher) |
| Transfected construct (hamster) | pCMV-HMGCR (TM1-8)-T7 | PMID:12535518 | | Membrane domain of hamster HMG CoA reductase inserted into pcDNA3 |
| Transfected construct (human) | siRNA against PDP1 (PDP1 siRNA-A) GGCGCGAGGUGCUGAUGAAUU | Dharmacon/Thermo Fisher Scientific | | |
| Transfected construct (human) | siRNA against PDP1 (PDP1 siRNA-B) GGCGCGAGGUGCUGAUGAAUU | Dharmacon/Thermo Fisher Scientific | ON-TARGET plus | |
| Transfected construct (human) | siRNA against PDP1 (PDP1 siRNA-C) CGACGUAGCUUUUGGCUUUUU | Dharmacon/Thermo Fisher Scientific | | |
| Transfected construct (human) | siRNA against PDP1 (PDP1 siRNA-D) AGUACAGCAUCGUGGACUAUU | Dharmacon/Thermo Fisher Scientific | | |
| Transfected construct (human) | siRNA against PDP1 (SMARTpool) (PDP1 siRNA-E) GCACAAUGUCACCGACGUA ACAUAAGCUUUCUCGUAUA GUAAACUGAACUUCGAGAA GCCCUGAUGUCGAGGUUA | Dharmacon/Thermo Fisher Scientific | ON-TARGETplus Human PLPP6 (403313) siRNA | |
| Transfected construct (jellyfish) | siRNA against GFP CAGCCACAACGUCUAUAUCUU | PMID:24025715 | | |
| Commercial assay or kit | Lipofectamine RNAiMAX | Thermo Fisher Scientific | Cat#13778500 | |
| Commercial assay or kit | X-tremeGENE HP DNA Transfection Reagent | Roche Diagnostics | Cat#45274000 | |
| Commercial assay or kit | Click-IT Protein Reaction Buffer Kit | Thermo Fisher Scientific | Cat#c10276 | |
| Commercial assay or kit | Novex WedgeWell 4%–20%, Tris-Glycine, 1.0 mm, Mini Protein Gel, 15-well | Thermo Fisher Scientific | Cat#c10276 | |
| Chemical compound, drug | Cholesterol | Sigma-Aldrich | Cat#C8667 | |
| Chemical compound, drug | Azido-geranylgeraniol | Thermo Fisher Scientific | Cat#C10249 | |
| Chemical compound, drug | Geranylgeraniol | Sigma-Aldrich | Cat#G3278 | |
| Chemical compound, drug | Menaquinone-4 | Sigma-Aldrich | Cat#809,896 | |
| Chemical compound, drug | Benzonase Nuclease | Sigma-Aldrich | Cat#E1014-5KU | |
| Chemical compound, drug | 25-Hydroxycholesterol | Avanti Polar Lipids | Cat#700019 P | |

*Continued on next page*

*Continued*

| Reagent type (species) or resource | Designation | Source or reference | Identifiers | Additional information |
|---|---|---|---|---|
| Chemical compound, drug | Trans, trans-farnesol | Sigma-Aldrich | Cat# 277,541 | |
| Chemical compound, drug | GGTI-298 | Cal Biochem | Cat# 345,883 | |
| Chemical compound, drug | MG-132 | Peptides | Cat# 12 L-3175-V | |
| Software, algorithm | Image J (Fiji) | NIH | | |

## Expression plasmids

The expression plasmid pCMV/TO-PDP1-Myc-FLAG encodes human PDP1 harboring C-terminal Myc and FLAG epitope tags. The cDNA for PDP1-Myc-FLAG was purchased from Origen (Rockville, MD) and cloned into the pcDNA4/TO vector (Thermo Fisher Scientific, Waltham, MA), which allows for transcription from the cytomegalovirus (CMV) promoter that can be repressed by the tetracycline repressor protein (TR). The plasmid pcDNA6/TR (Thermo Fisher Scientific) encodes TR under transcriptional control of the CMV promoter. The expression plasmid pCMV-PDP1 (S212T)-FLAG encodes human PDP1 harboring a C-terminal FLAG epitope and the catalytically inactivating S212T mutation. pCMV-HMGCR (TM1-8)-T7 encodes transmembrane domains 1–8 (amino acids 1–346) of hamster reductase with 3 copies of the T7 epitope at the C-terminus (*Sever et al., 2003a*).

## Cell culture

Monolayers of SV-589 cells, a line of immortalized human fibroblasts expressing the SV40 large T-antigen (*Yamamoto et al., 1984*), were maintained in medium A (DMEM containing 1000 mg/l glucose, 100 U/ml penicillin, and 100 mg/ml streptomycin sulfate) supplemented with 5 % (vol/vol) FCS at 37 °C, 5 % $CO_2$. UBIAD1-deficient SV-589 (ΔUBIAD1) cells (*Schumacher et al., 2018*) were grown in medium A supplemented with 10 % FCS and 1 mM mevalonate at 37 °C, 5 % $CO_2$. SV-589/PDP1-Myc-FLAG cells, which stably express tetracycline-inducible PDP1-Myc-FLAG, were generated as follows. We began by generating SV-589/TR cells, which stably express the tetracycline repressor protein (TR). On day 0, SV-589 cells were set up at $2.5 \times 10^5$ cells per 100 mm dish in medium A containing 5 % FCS. On day 1, cells were transfected with 6 µg/dish of pcDNA6/TR using FuGENE6 as described previously (*Schumacher et al., 2015*). After 16 hr at 37 °C, cells were switched to medium A supplemented with 5 % FCS and 5 µg/mL blasticidin. Fresh medium was added every 2–3 days until colonies formed after 2 weeks. Individual colonies were isolated using cloning cylinders and expression of TR was determined by immunoblot analysis. Cells from single colonies of cells expressing maximal levels of TR were selected and monolayers were maintained in medium A supplemented with 5 % FCS and 5 µg/mL blasticidin. Once obtained, SV-589/TR cells were set up at $2.5 \times 10^5$ cells per 100 mm dish in medium A supplemented with 5 % FCS and 5 µg/mL blasticidin. The next day, cells were transfected with pCMV/TO-PDP1-Myc-FLAG (1 µg/dish) using FuGENE6. Following incubation for 16 hr at 37 °C, cells were switched to medium A containing 5 µg/mL blasticidin and 500 µg/mL zeocin. Fresh medium was added every 2–3 days until colonies formed after 2 weeks. Individual colonies were isolated using cloning cylinders, and tetracycline-induced expression of PDP1-Myc-FLAG was determined by immunoblot analysis with anti-Myc and anti-FLAG antibodies. Single colonies of cells with maximal tetracycline-induced expression of PDP1 were selected and monolayers were maintained in medium A containing 5 µg/mL blasticidin and 500 µg/mL zeocin.

Cells were set up for transient transfection experiments on day 0 as described in figure legends. On day 1, triplicate dishes of cells were incubated with 3 µl of X-tremeGENE/µg DNA that was diluted in Opti-MEM I reduced serum medium (Thermo Scientific). Following incubation for 5 h at 37 °C, the cells received a direct addition of medium A supplemented with 10 % FCS (final concentration). On day 2, the cells were analyzed as described in the figure legends. To guard against potential genetic instability, an aliquot of each cell line was passaged for only 4–6 weeks, after which a fresh batch of

cells was thawed and propagated for subsequent experiments. All cell lines were confirmed to be free of mycoplasma contamination using the MycoAlert Mycoplasma Detection Kit (Lonza).

## RNA interference

RNA interference (RNAi) was performed as previously described with minor modifications. Duplexes of siRNAs were designed and synthesized by Dharmacon/Thermo Fisher Scientific. The siRNA duplex against GFP has been previously described (*Elsabrouty et al., 2013*); the sequence for siRNA duplexes targeting human PDP1 is provided in the 'Key Resources Table'. SV-589 cells were set up for experiments on day 0 as described in figure legends. On day 1, cells were incubated with 600 pmol of siRNA duplexes mixed with Lipofectamine RNAiMAX reagent (Thermo Fisher Scientific) diluted in Opti-MEM I reduced serum medium according to manufacturer's procedure. Following incubation for 6 h at 37 °C, cells received a direct addition of medium A containing 10 % FCS (final concentration). On day 2, cells were treated and analyzed as described in figure legends.

## Preparation of detergent-solubilized cell lysates, subcellular fractionation, and immunoblot analysis

SV-589 cells were set up for experiments on day 0 and subsequently treated as described in figure legends. Following incubations, triplicate dishes of cells for each variable were harvested and pooled for analysis. To prepare detergent-solubilized lysates, cells were resuspended in buffer containing SDS-lysis buffer (10 mM Tris-HCl, pH 6.8, 1 % (w/v) SDS, 100 mM NaCl, 1 mM EDTA, and 1 mM EGTA). Following passage through a 22-gauge needle 10–15 times, the samples were subjected to centrifugation at 20,000 X g. The resulting supernatant was designated the detergent-solubilized lysate. For subcellular fractionation by differential centrifugation, the cells were homogenized in buffer containing 10 mM HEPES-KOH, pH 7.6, 1.5 mM $MgCl_2$, 10 mM KCl, 5 mM EDTA, 5 mM EGTA, 250 mM sucrose, and a protease inhibitor cocktail consisting of 0.1 mM leupeptin, 5 mM dithiothreitol, 1 mM PMSF, 0.5 mM Pefabloc, 5 μg/ml pepstatin A, 25 μg/ml N-acetyl-leu-leu-norleucinal, and 10 μg/ml aprotinin. The homogenates were then passed through a 22-gauge needle 30 times and subjected to centrifugation at 1000 X g for 7 min at 4 °C. The 1000 X g pellet was resuspended in buffer (20 mM HEPES-KOH, pH 7.6, 2.5 % (v/v) glycerol, 0.42 M NaCl, 1.5 mM $MgCl_2$, 1 mM EDTA, 1 mM EGTA) supplemented with the protease inhibitor cocktail, rotated for 30 min at 4 °C, and centrifuged at 100,000 X g for 30 min at 4 °C. The supernatant from this spin was precipitated with acetone at –20 °C for at least 30 min; precipitated material was collected by centrifugation, resuspended in SDS-lysis buffer, and designated the nuclear extract fraction. The 1000 X g post-nuclear supernatant from the original spin was used to prepare the membrane fraction by centrifugation at 100,000 X g for 30 min at 4 °C. Each membrane fraction was resuspended in SDS-lysis buffer.

Protein concentration of detergent-solubilized lysates, nuclear extract, and membrane fractions were measured using the BCA Kit (Thermo Fisher Scientific). Prior to SDS-PAGE, aliquots of the nuclear extract, detergent lysates, and membrane fractions were mixed with the calculated amount of buffer containing 62.5 mM Tris-HCl, pH 6.8, 15 % (w/v) SDS, 8 M urea, 10 % (v/v) glycerol, and 100 mM DTT to obtain equal protein concentrations per μl of sample, after which 5 X SDS loading buffer was added to a final concentration of 1 X. Nuclear extract fractions were boiled for 5 min, whereas detergent-solubilized lysates and membrane fractions were incubated for 20 min at 37 °C prior to SDS-PAGE. After SDS-PAGE, proteins were transferred to Hybond C-Extra nitrocellulose filters (GE Healthcare, Piscataway, NJ). The filters were incubated with the antibodies described below and in the figure legends. Bound antibodies were visualized with peroxidase-conjugated, affinity-purified donkey anti-mouse, anti-rabbit, or anti-goat IgG (Jackson ImmunoResearch Laboratories, Inc, West Grove, PA) using the SuperSignal CL-HRP substrate system (Thermo Fisher Scientific) according to the manufacturer's instructions. Gels were calibrated with prestained molecular mass markers (Bio-Rad, Hercules, CA). Filters were exposed to film at room temperature. Antibodies used for immunoblotting to detect reductase (mouse monoclonal IgG-A9), SREBP-2 (rabbit monoclonal IgG-22D5), and UBIAD1 (mouse monoclonal IgG-1H12) were previously described (*Engelking et al., 2005*; *Jo et al., 2011*; *McFarlane et al., 2014*; *Schumacher et al., 2018*). Goat polyclonal anti-PDP1 IgG, mouse monoclonal IgG-26C4 (against RhoA), IgG-E6 (against Rap1), and IgG-B8 (against Cdc42) were obtained from Santa Cruz Biotechnology (Dallas, TX). Rabbit polyclonal anti-calnexin and anti-LSD1 IgG were purchased from

Novus Biologicals (Littleton, CO) and Cell Signaling (Beverly, MA), respectively. All antibodies were used at a final concentration of 1–5 µg/ml or 1:1000–5000 dilution.

## Immunofluorescence

SV-589 and SV-589/PDP1-Myc-FLAG cells were set up for experiments on day 0 and subsequently treated as described in figure legends. Following incubations, cells were washed with PBS, after which they were fixed and permeabilized for 15 min in methanol at –20 °C. Upon blocking with 1 mg/ml BSA in PBS, coverslips were incubated for 1 h at 37 °C with primary antibodies (mouse monoclonal IgG-1H2 against UBIAD1 and IgG-9E10 against c-Myc purified from the culture medium of hybridoma clone 9E10 (American Type Culture Collection, Manassas, VA)) diluted in PBS containing 1 mg/ml BSA. Bound antibodies were visualized with goat anti-mouse IgG conjugated to Alexa Fluor 488 (Thermo Fisher Scientific) as described in figure legends. Coverslips were also stained for 5 min with 300 nM 4',6-diamidino-2-phenylindole (DAPI) (Thermo Fisher Scientific) to visualize nuclei. The coverslips were then mounted in Mowiol 4–88 solution (Calbiochem/EMD Millipore, Billerica, MA) or Fluoromount G (Electron Microscopy Sciences, Hatfield, PA). Fluorescence imaging was performed using a Zeiss Axio Observer Epifluorescence microscope with a 63 x/1.4 oil Plan-Apochromat objective and Zeiss Axiocam color digital camera (Zeiss, Peabody, MA) in black&white mode as indicated in figure legends. Brightness levels were adjusted across entire images using ImageJ software (National Institution of Health, USA).

## Metabolic labeling with azido-geranylgeraniol and detection of azido-geranylgeranylated proteins

SV-589 and SV-589/PDP1-Myc-FLAG cells were set up for experiments on day 0 in medium A supplemented with 5 % FCS. On day one or two when cells reached at least 50 % confluency, cells were washed with PBS and refed medium A containing 5 % FCS and 10–30 µM azido-geranylgeraniol dissolved in DMSO. Following incubation for 24 hr at 37 °C, cells were harvested, washed with PBS, and pelleted by centrifugation at 500 X g. Cell pellets were then resuspended in buffer containing 50 mM Tris-HCl, pH 8.0, 1 % SDS, 250 U Benzonase endonuclease, and the protease inhibitor cocktail. Following incubation on ice for 15 min, the cells were homogenized by passage through a 22-gauge needle 10Xs, after which samples were clarified through centrifugation at 20,000 X g at 4 °C for 5 min. Clarified lysates were transferred to a new tube and an aliquot was removed for determination of protein concentration. Proteins in the remaining material were precipitated with methanol/chloroform/water; an appropriate amount of 50 mM Tris-HCl, pH 8.0 was used to solubilize precipitated material to achieve a final protein concentration of no more than 4 µg/ml. No more than 200 µg of solubilized protein in a total volume of 50 µl was labeled with biotin alkyne as described in the protocol for the Click-iT Protein Reaction Buffer Kit. Labeled samples were precipitated with methanol/chloroform/water, resolubilized in SDS-lysis buffer, and a small aliquot was removed for determination of protein concentration. The remaining sample was mixed with 5 X SDS loading buffer to achieve a final concentration of 1 X and subjected to SDS-PAGE. After SDS-PAGE, proteins were transferred to Hybond C-Extra nitrocellulose filters (GE Healthcare, Piscataway, NJ). Filters were blocked with PBS-Tween containing 1 % BSA, after which they were incubated with horseradish-conjugated streptavidin. Bound streptavidin was visualized using the SuperSignal CL-HRP substrate system (Thermo Fisher Scientific) according to the manufacturer's instructions. Filters were exposed to film at room temperature.

## in vitro vesicle-formation assay

SV-589 (ΔUBIAD1) cells were set up for experiments as described in the figure legends. Transient transfection of the cells with expression plasmids encoding Myc-tagged wild type and SCD-associated variants of UBIAD1 was carried out using FuGENE6 transfection reagent as described above. Following incubations described in figure legends, the cells were harvested, pooled, and subjected to subcellular fractionation; the in vitro vesicle-formation assay was carried out as described previously (*Schumacher et al., 2016*). In a volume of 100 µl, each reaction contained 50 mM HEPES-KOH (pH 7.2), 250 mM sorbitol, 70 mM potassium acetate, 5 mM potassium EGTA, 2.5 mM magnesium acetate, 1.5 mM ATP, 0.5 mM GTP, 10 mM creatine phosphate, 4 units/ml creatine kinase, protease inhibitors, 80–100 µg protein of microsomes obtained from transfected SV-589 (ΔUBIAD1) cells, and 10 µg cytosol obtained

from isoprenoid-depleted, reductase-deficient UT-2 cells (*Mosley et al., 1983*). Reactions were carried out in siliconized 1.5 ml microfuge tubes for 20 min at 37 °C, after which they were terminated and subjected to centrifugation at 16,000 g for 3 min at 4 °C to obtain P16 pellet and S16 supernatant fractions. The S16 fraction was next subjected to centrifugation at 100,000 g; the pellet of this spin was designated P100. The P16 and P100 fractions were resuspended in SDS-lysis buffer, mixed with SDS-PAGE loading buffer, and incubated at 95 °C for 5 min. Aliquots of P16 and P100 fractions, referred to membrane and vesicles, respectively, were subjected to SDS-PAGE, transferred to nylon filters, and analyzed by immunoblot using IgG-9E10 against Myc to detect transfected UBIAD1.

## Synthesis of [³H]menaquinone-4 (MK4) in control and PDP1 knockdown cells

In studies measuring incorporation of [³H]menadione into MK-4, cells were set up on day 0 and transfected with siRNA duplexes against GFP and PDP1 as described in figure legends. Following siRNA transfection, cells were refed medium A containing 5 % FCS and 2.5 µCi/ml [³H]menadione (cold menadione was added to achieve a final concentration of 50 nM). After 16 hr at 37 °C, cells were washed twice with PBS + 2 % BSA, followed by an additional wash with PBS. The cells were then lysed with 0.1 N NaOH; the resulting lysates were mixed with recovery solution containing 16 µg/ml MK-4, 0.025 µCi/ml [¹⁴C]cholesterol, and 16 µg/ml unlabeled cholesterol and extracted with dicholoromethane:methanol (2:1). The lipids were dried down, resuspended in dicholoromethane, and spotted on thin layer chromatography (TLC) plates that were developed in chloroform. The lipids were visualized by staining in iodine vapor and incorporation of [³H]menadione into MK-4 was determined by scintillation counting. The values were corrected for recovery as judged by the percent of [¹⁴C]cholesterol recovered in each sample. An aliquot of each sample was taken for protein determination.

## Reproducibility of data

All results were confirmed in at least two independent experiments conducted on different days using different batches of cells.

## Acknowledgements

We thank Drs. Xiaochun Li and Marc Schumacher for critical review of the manuscript and Lisa Beatty, Ijeoma Dukes, and Camille Harry for help with tissue cutlure. This work was supported by National Institutes of Health grant HL-20948 (RD-B).

## Additional information

### Funding

| Funder | Grant reference number | Author |
| --- | --- | --- |
| National Institutes of Health | HL-20948 | Russell A DeBose-Boyd |
| National Institutes of Health | GM-144039 | Russell A DeBose-Boyd |

The funders had no role in study design, data collection and interpretation, or the decision to submit the work for publication.

### Author contributions

Rania Elsabrouty, Conceptualization, Data curation, Investigation, Methodology, Visualization, Writing – original draft, Writing – review and editing; Youngah Jo, Seonghwan Hwang, Dong-Jae Jun, Investigation, Methodology, Writing – review and editing; Russell A DeBose-Boyd, Conceptualization, Data curation, Funding acquisition, Investigation, Methodology, Visualization, Writing – original draft, Writing – review and editing

### Author ORCIDs

Youngah Jo http://orcid.org/0000-0001-6779-3891

Russell A DeBose-Boyd [ID] http://orcid.org/0000-0002-7452-5227

**Decision letter and Author response**
Decision letter https://doi.org/10.7554/eLife.64688.sa1
Author response https://doi.org/10.7554/eLife.64688.sa2

## Additional files

### Supplementary files
- Transparent reporting form
- Source data 1. Source data for *Figures 1–6*.

### Data availability
All data generated or analyzed during this study are included in the manuscript.

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
