## [Editor Report]

This manuscript investigates the regulation of the mevalonate pathway by geranylgeraniol. In a series of elegant and convincing experiments, the authors show that modulating the expression levels of Polyisoprenoid Diphosphate Phosphatase (PDP1) alters endoplasmic reticulum-associated protein degradation of HMGCoA reductase though UBAID1. They also show that modulation of geranylgeraniol levels through exogenous addition or depletion of PDP1 alters stability and levels of important small GTPases, such as RhoA. The biology here is fascinating and very important, not only as it is pertinent to a fundamental mechanism of biological control but also as directly related to cholesterol-lowering statin therapy.

---

## [Decision Letter]

**Decision letter after peer review:**

Thank you for submitting your article "Type 1 Polyisoprenoid Diphosphate Phosphatase Modulates Geranylgeranyl-Mediated Control of HMG CoA Reductase and UBIAD1" for consideration by *eLife*. Your article has been reviewed by 3 peer reviewers, and the evaluation has been overseen by Suzanne Pfeffer as the Senior and Reviewing Editor. The following individual involved in review of your submission has agreed to reveal their identity: Tobias C Walther (Reviewer #2).

Below you will find a set of recommendations, which, if completed, will lead to publication in *eLife*.

Essential revisions:

1. In the proposed model, UBIAD1 ER-Golgi trafficking is controlled by GGpp levels. However, the data cannot exclude an enzymatic-independent function of PDP1 in retaining UBIAD1 at the ER. This can be addressed by including a control cell line expressing catalytically inactive PDP1. Along the same line, it would be important to show that UBIAD1 mutants refractory to GGpp levels (for example N102S) restores HMGCR levels upon PDP1 depletion. These experiments would strengthen the role of GGpp in controlling UBIAD1 dynamics.

2. Some of the changes in protein degradation kinetics are modest and their interpretation is complicated by different basal steady state levels of HMGCR. Ideally pulse-chase experiments should be used in these cases (Figures 1B, 1C and 3B) and quantification from several independent experiments should be provided.

To aid the review process, we include the additional detailed comments from the reviewers here.*Reviewer #1:*

1) Showing several key results with a catalytic mutant of PDP1 would strengthen their conclusions. Or, alternatively by actually measuring GGpp/GGOH levels to show real changes in the balance (mass spec or TLC). It seems like there a chance that PDP1 is only serving a chaperone-like function for UBIAD1/HMGCR, or possibly serving as a sink for GGpp or GGOH?

2) Figure 2B, it seems like this experiment performed in parallel with membranes from cells overexpressing myc-PDP1-Flag (particularly by adding a catalytic mutant) would provide stronger supporting evidence for the authors' model. Alternatively, GGOH supplementation would help support this.

*Reviewer #2:*

This manuscript by the DeBose-Boyd group investigates the regulation of the mevalonate pathway by geranylgeraniol (GGOH), which is generated by the phosphatase PDP1.

In a series of elegant and convincing experiments, the authors show that modulating the expression levels of PDP1 alters ERAD of HMGCoA reductase though UBAID1, and controls localization of UBIAD. They also show that modulation GGOH levels through exogenous addition of depletion of PDP1 alters stability and levels of important small GTPases, such as RhoA.

The biology here is fascinating and very important, not only as it is pertinent to a fundamental mechanism of biological control but also as directly related to cholesterol-lowering statin therapy. This paper is exceptionally well done, every figure is convincing and the data appear "solid as a rock". All the remains, are some curious questions that the authors might consider for the discussion.

• Is PDP1 constitutively active or regulated under some conditions? What determines the levels of GGOH? The kinase reaction or the phosphatase reaction? Are there mutations in human PDP1 reported?

• Are there cellular consequences to the stabilization of RhoA and other cytoskeleton regulating GTPases?

• Is there a possibility that not only forward but also retrograde (e.g., COPI-mediated) trafficking of UBIAD is regulated?

---

## [Author Response]

Essential revisions:1. In the proposed model, UBIAD1 ER-Golgi trafficking is controlled by GGpp levels. However, the data cannot exclude an enzymatic-independent function of PDP1 in retaining UBIAD1 at the ER. This can be addressed by including a control cell line expressing catalytically inactive PDP1. Along the same line, it would be important to show that UBIAD1 mutants refractory to GGpp levels (for example N102S) restores HMGCR levels upon PDP1 depletion. These experiments would strengthen the role of GGpp in controlling UBIAD1 dynamics.

The reviewer raises an excellent point and suggests a great experiment. In the revised manuscript we provide two experiments. Figure 1 shows that overexpression of PDP1 blocked GGOH-induced ERAD of HMGCR, indicating the phosphatase hydrolyzes pools of GGOH-derived GGpp that control HMGCR ERAD. We now show in Figure Supplement-1 that overexpression of catalytically inactive PDP1 (S212T) enhanced ERAD of the membrane domain of HMGCR, which is necessary and sufficient for the reaction. Figure 2 shows that overexpression of PDP1 prevents the GGOH-induced translocation of UBIAD1 from the ER to the Golgi, suggesting that the phosphatase hydrolyzes pools of GGOH-derived GGpp that modulate ER-to-Golgi transport of UBIAD1. We now show in Figure 2—figure supplement-1 that UBIAD1-Myc localizes to the Golgi of isoprenoid replete cells. The HMGCR inhibitor compactin, which depletes cells of nonsterol isoprenoids, triggers the relocation of UBIAD1 from the Golgi to ER as we previously reported. UBIAD1-Myc continued to localize to the Golgi of isoprenoid-replete cells upon overexpression of catalytically inactive PDP1 (S212T); however, the protein remains in the Golgi when cells are subjected to compactin treatment. Together, these findings rule out an enzymatic-independent function of PDP1 in modulation of UBIAD1 transport and HMGCR ERAD.

2. Some of the changes in protein degradation kinetics are modest and their interpretation is complicated by different basal steady state levels of HMGCR. Ideally pulse-chase experiments should be used in these cases (Figures 1B, 1C and 3B) and quantification from several independent experiments should be provided.

We now show a cycloheximide chase study to analyze ERAD of HMGCR in control and PDP1 knockdown cells (Figure 3—figure supplement 1A). The results are similar to those observed in Figure 3B; knockdown of PDP1 enhanced GGOH-induced ERAD of HMGCR. We also provide another experiment in Figure 3—figure supplement 1B that shows GGOH-induced ERAD of HMGCR is blocked by the proteasome inhibitor MG-132. Considering results from Figure 3A, which shows the disappearance of HMGCR upon PDP1 knockdown, Figure 4C, 4E, and 6A showing that PDP1 knockdown blunts compactin-induced stabilization of HMGCR, we feel confident in our conclusion that PDP1 governs pools of GGpp that augment the ERAD of HMGCR.

Reviewer #2:This manuscript by the DeBose-Boyd group investigates the regulation of the mevalonate pathway by geranylgeraniol (GGOH), which is generated by the phosphatase PDP1.In a series of elegant and convincing experiments, the authors show that modulating the expression levels of PDP1 alters ERAD of HMGCoA reductase though UBAID1, and controls localization of UBIAD. They also show that modulation GGOH levels through exogenous addition of depletion of PDP1 alters stability and levels of important small GTPases, such as RhoA.The biology here is fascinating and very important, not only as it is pertinent to a fundamental mechanism of biological control but also as directly related to cholesterol-lowering statin therapy. This paper is exceptionally well done, every figure is convincing and the data appear "solid as a rock". All the remains, are some curious questions that the authors might consider for the discussion.• Is PDP1 constitutively active or regulated under some conditions? What determines the levels of GGOH? The kinase reaction or the phosphatase reaction? Are there mutations in human PDP1 reported?

We are not aware of conditions under which PDP1 is regulated; however, this has not been examined in detail.

• Are there cellular consequences to the stabilization of RhoA and other cytoskeleton regulating GTPases?

This is a possibility that we have not addressed.

• Is there a possibility that not only forward but also retrograde (e.g., COPI-mediated) trafficking of UBIAD is regulated?

We have not obtained evidence that retrograde transport of UBIAD1 is regulated. In a previous study, we showed that UBIAD1 cycles continuously between the Golgi and ER and becomes trapped in the ER when GGpp is depleted. This was indicated by the finding that blocking export from the ER results in the relocalization of UBIAD1 to the ER, even when GGpp is abundant (Schumacher et al. 2016 JLR 57:1286)